# 21ˢᵗ-century stagnation in unvegetated sand-sea activity

Andrew Gunn[1,2], Amy East[3] & Douglas J. Jerolmack [2,4 ✉]

Sand seas are vast expanses of Earth's surface containing large areas of aeolian dunes—topographic patterns manifest from above-threshold winds and a supply of loose sand. Predictions of the role of future climate change for sand-sea activity are sparse and contradictory. Here we examine the impact of climate on all of Earth's presently-unvegetated sand seas, using ensemble runs of an Earth System Model for historical and future Shared Socioeconomic Pathway (SSP) scenarios. We find that almost all of the sand seas decrease in activity relative to present-day and industrial-onset for all future SSP scenarios, largely due to more intermittent sand-transport events. An increase in event wait-times and decrease in sand transport is conducive to vegetation growth. We expect dune-forming winds will become more unimodal, and produce larger incipient wavelengths, due to weaker and more seasonal winds. Our results indicate that these qualitative changes in Earth's deserts cannot be mitigated.

[1] School of Earth Amtosphere and Environment, Monash University, Clayton, Australia. [2] Department of Earth and Environmental Sciences, University of Pennsylvania, Philadelphia, PA, USA. [3] Pacific Coastal & Marine Science Center, United States Geological Survey, Santa Cruz, CA, USA. [4] Department of Mechanical Engineering and Applied Mechanics, University of Pennsylvania, Philadelphia, PA, USA. ✉email: sediment@sas.upenn.edu

Sand seas are some of the least hospitable domains of Earth's surface; the atmosphere is dry and windy with extreme diurnal cycles[1] and the land is barren and erodible[2]. Spanning 100–600,000 km², sand seas (or ergs) host the largest expanses of repeating patterned topography on the planet, dune fields, which have morphology linked to the geologically controlled supply of sand grains and the persistence and direction of sand-transporting winds often tied to the seasons[3,4]. Three fundamental properties of sand seas make them landscapes with exceptional sensitivity to climate: first, even under constant climatic and geological conditions, regions with dunes never reach an equilibrium state and instead coarsen indefinitely[5]; second, unlike networked landscapes such as river basins, the parts of these landscapes dominated by loose sand, when stressed by unconfined flow, are highly susceptible to erosion wherever unconsolidated sediment occurs; and third, sand is only transported by winds that exceed a threshold speed, and since this threshold condition is frequently met (at least on seasonal or shorter time scales) the landscape is persistently in a near-critical condition[6]. These final two points imply that sand seas are exquisitely sensitive to small changes to the tails of wind-speed distributions. Furthermore, the activity of sand seas—i.e. the amount of landscape change by sediment transport—scales nonlinearly with the wind speed in excess of threshold[7]. The threshold is principally set by precipitation, both directly via liquid capillary bridges between sand grains and indirectly through vegetation[8–10]. Increasing wind and precipitation therefore have opposing effects on sand-sea activity. Importantly vegetation introduces cusp catastrophe in sand-sea dynamics: once activity stagnates below some threshold such that vegetation can take root on unvegetated dunes, activity must exceed a far higher threshold in order to return to an unvegetated state[11,12]. This represents a regional tipping point in the state of an arid landscape. Previous studies have focused primarily on regions where dunes are now partly stabilized by vegetation (i.e. the Kalahari Desert[13]), concluding that in a warmer climate a lower ratio of precipitation to potential evapotranspiration would decrease vegetation enough to reactivate some dune fields[11–15].

Here we focus on how contemporary climate change may impact currently active, unvegetated sand seas. Using the European Consortium coupled Earth System Model (ESM), EC-Earth3 (Methods)[16], we examine ensembles of ESM runs for historical (1850-2014) and Tier-1 SSP scenarios (2015-2100) computed for the recent Climate Model Intercomparison Project (CMIP6; Shared Socioeconomic Pathways, SSPs, are trajectories of global socioeconomic and technological development projected to respond to and potentially mitigate climate change)[17,18]. We pair aeolian sediment-transport theory with 3-hourly fields of precipitation flux and 10-m wind vectors to calculate sand activity for all ($n = 45$) of Earth's active sand seas[7,19] (Figs. 1a and S1, Table S1, and Methods). An example for the Grand Erg Occidental in northern Algeria is shown in Fig. 1b–d. We find that almost all currently active sand seas are predicted to become less active under all future SSP scenarios—even those with significant anthropogenic mitigation strategies—implying that the impact of past human action cannot be reversed but that its magnitude can be modulated. By considering the tails of activity distributions, we highlight some second-order impacts of sand-sea stagnation specific to the morphology of dunes and sand-transport events, finding that both are strongly linked to seasonality in most sand seas.

## Results

**Sand-sea activity**. First we examine the global trend in sand transport through time as predicted by the EC-Earth3 ESM. Atmospheric fields on the nominally 100-km grid of the model are

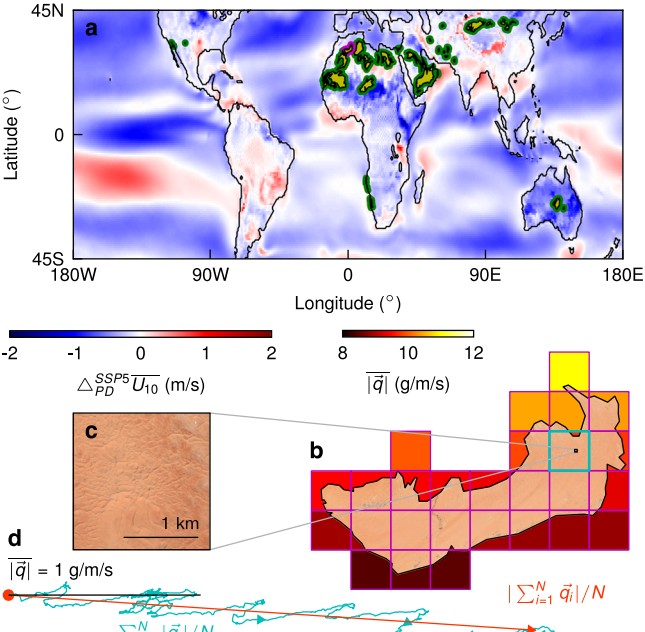

**Fig. 1 Sand-sea locations and flux extraction example. a** 45 sand seas (yellow, with thick border for clarity; green except for purple example in (**b**)) analysed in this study on an ensemble-average map of the annual-average 10-m wind speed anomaly from present-day (decade ending 2014) to the predicted SSP5-8.5 decade ending 2100, $\triangle_{PD}^{SSP5}\overline{U_{10}}$ (m/s). **b** A LANDSAT image of an example sand sea, the Grand Erg Occidental, northern Algeria, overlaying the nominally 100-km ESM grid (purple) showing the ensemble-average present-day annual sand flux magnitude $|\vec{q}|$ (g/m/s). **c** A MAXAR image of dune morphology in the cyan tile. (**d**) An example sand flux trajectory (cyan) for one ensemble member of the cyan tile in (**b**) for the 2005-2014 decade with a scale $|\vec{q}| = 1$ g/m/s (black line); the length of the orange and cyan lines give the resultant $|\sum_{i=1}^{N} \vec{q_i}|/N$ and total $\sum_{i=1}^{N} |\vec{q_i}|/N$ sand flux magnitudes, respectively, where $N$ is the number of samples.

filtered spatially by using sand-sea masks manually extracted from LANDSAT imagery and weighting the grid tiles according to their coverage of the sand sea[1] (Methods), allowing us to find the average sand flux for each sand sea (Methods). Then a global time series for each ensemble member in a scenario is found as the sand-sea area-weighted average sand flux. We plot the mean global average sand flux time series smoothed over a 5-year window shadowed by the ensemble standard deviation for each scenario (Fig. 2a). A clear and significant trend of a future global decrease in sand flux in the sand seas emerges from the forcing variability in time (noise in the average), and intrinsic variability in the climate system (width of the shadow). The magnitude of the mean tendency in each future time series goes monotonically with scenario radiative forcing. We find no clear mean trend in the historical time series relative to the SSP scenarios, and note that due to the global distribution of sand seas and the 5-year smoothing in Fig. 2a, climate modes or seasonality in a given sand sea's flux signal are not apparent in the globally averaged time series.

The smoothed time series does not reflect the bursty, nonlinear behavior of aeolian sediment transport[20]. An example for a particularly severe sand storm in the Namib Sand Sea in Fig. 2b shows that the EC-Earth3 sand flux time series can also be viewed as a set of discrete events of size $Q = \Delta t \sum_{i=1}^{N} q_i$ (kg/m), where $i$ is the index of measurements of stepsize $\Delta t$ (3 hours) that lasts for $N$ steps, between wait-times, $T$. Wait-times—i.e. times of inactivity between transport events—are defined as $T = M\Delta t$ (s), where

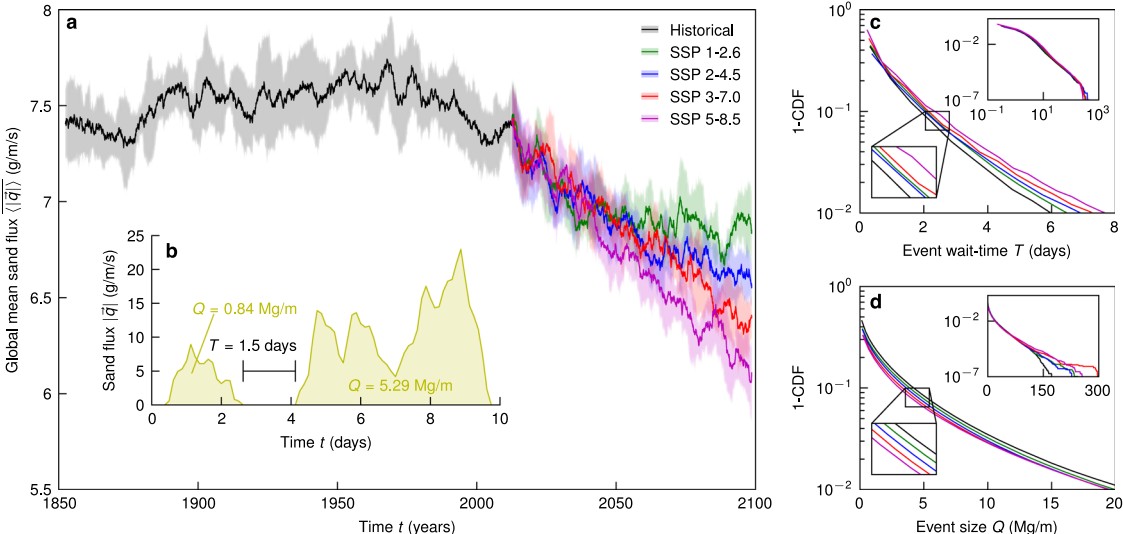

**Fig. 2 Historical and SSP global sand flux activity. a** Modelled time series of the 5-year smoothed globally-averaged sand flux magnitude $\langle |\overrightarrow{q}| \rangle$ (g/m/s) for the historical (black) and future SSP (1–2.6, green; 2–4.5, blue; 3–7.0, red; 5–8.5, purple) scenarios; ensemble mean (lines) and ±1 standard deviation (shaded envelopes) are shown. **b** An example sand flux magnitude $|\overrightarrow{q}|$ (g/m/s) time series (yellow) from the Namib Sand Sea for one tile in one ensemble member defining the event size $Q$ (Mg/m) (shaded yellow areas) and wait-time $T$ (days) (horizontal black line). The global Magnitude-Frequency plot for each scenario (lines colored as in (**a**)) of (**c**) $T$ and (**d**) $Q$ up to the 99th percentile with insets top-right showing the full CDFs to the $(100 - 10^{-5})$th percentile and bottom-left to compare scenarios at the 90th percentile.

there are $M$ inactive steps. This perspective is useful when considering extreme events and the duration of inactivity, both of which are relevant in the ability for vegetation to take hold. We plot the global Magnitude-Frequency distributions (1-CDFs) of $Q$ and $T$ for the final decade in each scenario (2005-2014, historical; 2091-2100, SSP) and find significant changes with radiative forcing (Fig. 2c and d). Magnitude-Frequency plots for both variables have fat tails and are approximately Poissonian with inflation at short times (Fig. S2), likely owing in part to the finite timestep. There is a clear trend of decreasing likelihood of extreme events and increasing likelihood of long periods without transport with increasing future radiative forcing relative to 2005-2014 (Fig. 2c and d), both conducive to increased opportunity for ecological growth[2,11]. The tails of these CDFs can be represented simply with a single parameter by the 99$^{th}$ percentile event size $Q_{99}$ (Mg/m) and wait-time $T_{99}$ (days).

Next we break down the global trend to view the percent relative change in individual sand sea flux magnitude from the present-day decade to 2091–2100 in the SSP scenarios (Fig. 3a). The predicted global stagnation is principally borne out in the northern hemisphere, which has significantly more sand-sea area. The southern hemisphere sand seas in central Australia and southern Africa instead see a moderate increase in activity, which is qualitatively consistent with previous studies[13,21,22] (Fig. S3a). Despite this hemispheric contrast, we find that across all but the smallest sand sea in this study, White Sands Dune Field, the rare event wait-times $T_{99}$ (days) are predicted to increase in the future, particularly for the Sinai Negev Erg, An Nafud and Ad Dahna sand seas (Figs. 3b and S3c). The increase in southern African sand-sea activity on the Atlantic coastline can be attributed in part to a relatively large increase in extreme event sizes $Q_{99}$ (Mg/m) (Figs. 3c and S3d). Comparing Fig. 3b and c, we see that changes in mean flux $|\overrightarrow{q}|$ are manifest predominately from longer periods of quiescence rather than from decreased severity of flux events.

**Dune morphology**. Sand-flux magnitude is a useful measure for sand sea activity, dust emission, and as a rate parameter for dune coarsening, but it is not sufficient to determine dune

morphology[23]. The principle dune forms—barchan, transverse, linear and star—arise under unimodal, unimodal, bimodal and multimodal sand flux direction regimes, respectively, with the former two being delineated by low and high sand-supply states, respectively[5,23,24] (Fig. 1c). As climate changes sand-flux magnitudes, directional regimes of sand flux may change too. This could lead to new dune morphology or perhaps superimposing new forms upon present giant dunes[25]. Our forecast window of a century is short compared to the timescales of the evolution of the world's large dunes (millennia)[2]; therefore, climatically induced changes in wind regime are unlikely to erase the landscape's memory of historical forcing. However, a century is enough time to produce the incipient, smallest-scale dunes in the landscape— on the scale of tens of meters—from which all larger dunes subsequently coarsen in a pattern-reformation process[26].

First we can assess changes in the wavelength of these incipient dunes, which arise from a hydrodynamic instability between the near-surface winds and the topography that they rework. Through linear stability analysis, that has been validated in the field and laboratory[26–28], the wavelength $\lambda_c$ (m) of incipient dunes is a function of the inverse square of mean wind in excess of threshold shear velocity, $\lambda_c \sim \overline{u_*}^{-2}|_{u_* > u_{*,cr}}$ (Methods). It is therefore not sensitive to longer periods of inactivity, but rather weakened activity. As the scaling suggests, we see the most future change in $\lambda_c$ for sand seas that have weaker dune-forming winds (Fig. 3g), such as those in east Asia. In most cases the EC-Earth3 ESM predicts incipient dunes will grow in wavelength because winds weaken, with changes on the order of the dune wavelengths themselves, sometimes in excess of 10 meters (Fig. S3e).

In Fig. 3d we plot the percent relative change from the decades 2005-2014 to 2091-2100 in the resultant sand flux magnitude for each Tier-1 SSP scenario as predicted by the EC-Earth3 ESM. The resultant sand flux magnitude $|\sum\overrightarrow{q}|$ is necessarily less than the absolute sand flux magnitude $\sum|\overrightarrow{q}|$ (Fig. 1d), and drives dune migration[24]. We see more variance in resultant flux changes across the sand seas and scenario cases than for absolute flux, owing to certain flux-contributing wind modes weakening more than others. We then investigate the ratio of the resultant to absolute flux

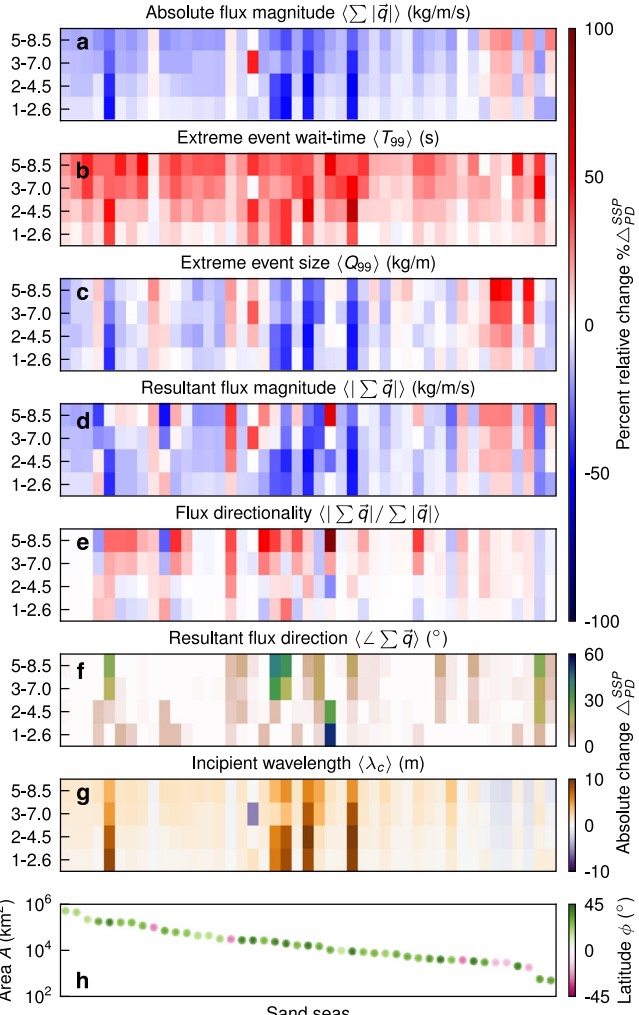

Fig. 3 Changes in key variables for dune morphology. Given for each sand sea in descending-area order (horizontally) from the present-day decade 2005–2014 to future decade 2091–2100 for each SSP scenario in ascending-radiative forcing order (vertically) are; percentage relative changes $\%\Delta_{PD}^{SSP}$ for (a) total sand flux magnitude $\sum|\vec{q}|$ (kg/m/s), 99th percentile event (b) wait-time $T_{99}$ (s) and (c) size $Q_{99}$ (kg/m), (d) resultant sand flux magnitude $|\sum\vec{q}|$ (kg/m/s), (e) flux directionality $|\sum\vec{q}|/\sum|\vec{q}|$, and absolute changes $\Delta_{PD}^{SSP}$ in (f) resultant flux direction $\angle\sum\vec{q}$ (°) and (g) incipient dune wavelength $\lambda_c$ (m). (h) Sand-sea area $A$ (km$^2$) colored by centroid latitude $\phi$ (°).

magnitudes, what we term 'flux directionality', which is a measure between 0 and 1 that indicates net-zero flux and purely unidirectional flux, respectively (Figs. 3e and S3b). Flux directionality increases in the future in most cases, particularly in subtropical Africa (Fig. S1), signalling that the decrease in sand-sea activity is predominantly occurring in directions of less flux. This also causes the resultant flux vector direction to change with its magnitude too (Fig. 3f), which for high-mobility sand seas (i.e. those with high flux directionality) implies that dunes may start migrating in a different direction. One example to highlight is the Namib Sand Sea, estimated to be 1 My old that is currently covered by a mixture of giant linear and star dunes[29], which is predicted to see a shift from moderate to high flux directionality and an associated veering of resultant flux direction of around 20°, due largely to an increase in flux event size in the windy season (Fig. S3).

The morphology of dunes is largely dictated by the seasonality of winds over a sand sea[2]. Quantifying the seasonality of sand-sea

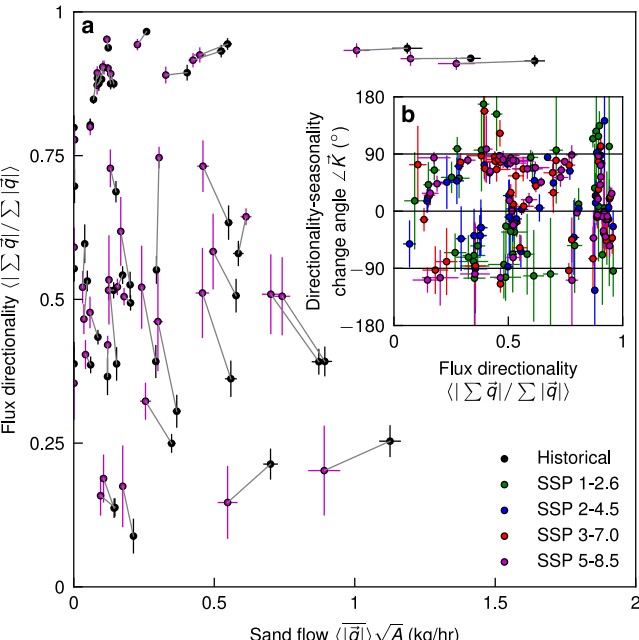

Fig. 4 Seasonality and slower, unidirectional dunes. a Sand flow, the product of a sand sea's flux $\langle|\vec{q}|\rangle$ (kg/m/hr) and average width $\sqrt{A}$ (m), against flux directionality $\langle|\sum\vec{q}|/\sum|\vec{q}|\rangle$ for all sand seas in decadal averages for 2005–2014 (black dots) and 2091–2100 in the highest radiative forcing case SSP 5–8.5 (purple dots) linked for each sand sea with grey vectors. b Flux directionality against the angle of vectors $\angle\vec{K}$ (°) in Figure S5: $\angle\vec{K}$ is the sensitivity of change in the flux directionality $\langle|\sum\vec{q}|/\sum|\vec{q}|\rangle$ to change in the flux seasonality from 2005–2014 to 2091–2100 in each SSP scenario (different colors denoted in the legend). If $\angle\vec{K}>0°$ then flux directionality increases in the future, and if $-90°>\angle\vec{K}>90°$ then seasonality increases in the future. A sand sea's flux seasonality is defined as the proportion of annual flux $\sum\langle|\vec{q}|\rangle$ (kg/m/s) that occurs during the quarter (consecutive 3-month period) of the year with the most flux, $\max\{\langle|\vec{q}|\rangle\}$ (kg/m/s). All dots have ±1 ensemble standard deviation error bars.

activity as the proportion of the annual activity that occurs during the most active quarter of the year we see that, aside from the unimodal tropical sand seas in Africa and Middle East owing to the persistence and strength of trade winds (Fig. S4), flux directionality and seasonality are correlated (Fig. S5). Indeed, future changes in seasonality are also predicted to follow this correlation and are larger for increased radiative forcing SSP scenarios (Fig. S5). Scaling sand flux by sand-sea length—i.e. sand flow (kg/s)—we see that decreasing sand flow through weakening winds is associated with increased flux directionality (Fig. 4a). The attractor in the top left corner of Fig. 4a represents a global transition toward unimodal dunes of weakened flux in sand seas. The sensitivity of future changes in flux directionality to seasonality, quantified by the angle of the coeval change vector

$$\angle\vec{K} = \arctan 2\left(\Delta_{PD}^{SSP}\langle|\sum\vec{q}|/\sum|\vec{q}|\rangle, \Delta_{PD}^{SSP}\max\{\langle|\vec{q}|\rangle\}/\sum\langle|\vec{q}|\rangle\right)$$

(Fig. 4b), is majority between 0° and 90°. This indicates that weakening winds are predicted to affect sand flux most outside of the most active season of sand-sea activity.

## Discussion

Under the CMIP6 Tier-1 SSP scenarios[17], the EC-Earth3 ESM predicts that human-induced climate change will cause a global stagnation in sand-sea activity during the 21st-century, regardless of future actions, which could be clearly identifiable through

natural variability by 2100. This change can mostly be attributed to changes in wind rather than precipitation (Fig. S6). Since sand transport is a close-to-threshold phenomenon, the increase in the amount of time of inactivity is more significant than the weakening of flux event size (Fig. 3b and c), and the interplay of the threshold and seasonality is predicted to lead to more unidirectional sand seas (Fig. 4).

Overall the stagnation may lead to the rise in the vegetation of certain presently unvegetated sand seas that would represent a tipping point[10], and may decrease the contributions of some source areas to the global dust budget[30,31], although dust sourcing from dry lake beds would continue. Interestingly, the interplay of vegetation and flux direction may lead to increasing prevalence of parabolic dunes[9,32,33]. However, in many of Earth's hyper-arid landscapes, including some of the 45 sand seas studied here, the principle bottleneck for the rise of vegetation (and therefore parabolic dunes) is not insurmountable wind power but a lack precipitation[34]. Our prediction of prevailing stagnation and associated precipitation increase (Fig. S6) is inconsistent at the global scale with the consensus of previous regional studies which predict that vegetation loss in a warmer climate will lead to reactivation of currently stable dune fields[12–14]. While we do not focus on those cases, we note that the wind strength changes seen in the ESM around the partially-vegetated Thar Desert, and coastal dunes in north Chile and south Peru, warrant further study of potential short-term reactivation (Fig. 1a). Increased vegetation would affect the regional carbon cycle and potentially increase atmospheric $CO_2$ drawdown, though likely to a modest degree. We believe our results, which are broadly consistent with a CMIP6 ESM-ensemble ($n = 24$, Fig. S7) but should be validated further when possible, are an indication that large-scale change detection in presently-unvegetated sand seas may be a potentially useful signal of indirect human-induced changes to Earth's geomorphology in the Anthropocene. This could be achieved with remote sensing tools, such as ICESat-2 or CubeSats, that can resolve both vegetation and incipient dunes—the building-blocks of sand-sea topography that have the least memory of past climate. The results here contribute to a growing understanding of how humans are not only affecting Earth's surface through direct land-use change[15,35,36], but indirectly through the inertia of climate[37].

## Methods

**EC-Earth3 ESM**. The European Consortium (EC) Earth System Model (ESM), EC-Earth3, is one of the ESMs used to perform a suite of simulations within, and consistent with, the Coupled Model Intercomparison Project 6 (CMIP6)[16,18]. The simulations we focus on in this paper are forced by; reanalysis of observational data for 1850-2014, in the 'historical' scenario[38], and hypothetical future greenhouse gas emission and human-activity scenarios ($N = \{1, 2, 3, 5\}$) agreed under peer-consensus that create approximate radiative forcing values ($F = \{2.6, 4.5, 7.0, 8.5\}$ W/m²) through Shared Socioeconomic Pathways (SSP$N$-$F$) in the period 2015-2100[17,39]. The four future scenarios we analyse are termed 'Tier-1'[17].

EC-Earth3 is the CMIP6 ESM we focus on because it is the only one that currently has public data that satisfy all of the following criteria. It has a grid resolution equal to or below 100 km (nominally) in order to capture all sand seas reasonably, has 3-hourly data for 10-m wind and precipitation, and has multiple ensemble members for all four Tier 1 SSP scenarios and the historical scenario. Fortunately, it is consistent with most other CMIP6 ESMs in average changes in wind speed (Fig. S7), and therefore also represents a faithful 'best estimate' from the CMIP6 group to focus on.

We do not discuss the details of the model here, as it is a fully-coupled ESM with many aspects that contribute to the wind and precipitation[16,40]. Wind and precipitation are an expression of the coupled interactions between the atmosphere module of the ESM with the other modules, such as the ocean and ice modules. Of principle interest to near-surface winds relevant to sand transport in the ESM is the planetary boundary layer scheme—which transfers momentum from the free atmosphere to the land—since this scheme incorporates the role of surface heat fluxes into the transfer of momentum, and sand seas have extreme surface heat fluxes[1].

**Sand-sea masking**. Masks of 45 sand seas were drawn manually using Google Earth over LANDSAT imagery[1]. These are defined as regions of erodible sand with

active dunes void of vegetation that have a continuous and singular boundary. Sand-sea areas are calculated from the projection of these masks onto the local UTM into units of meters. For all CMIP6 ESMs, the same method illustrated in Fig. 1b is used to find the relevant grid points in a given ESM for the atmospheric fields over a given sand sea. The contribution of calculated sand flux vectors from each grid point to the average for a sand sea (and subsequently to the area-averaged global value) is based on the proportion of the grid-point tile's area covered by the sand sea. The only exception to this is the trivial case when the entire sand-sea lies within one grid-point tile (which does not occur for any sand sea in the EC-Earth3 ESM grid, but does for some coarser gridded ESMs in the CMIP6 ensemble). The globally-averaged value is then the area-weighted average of all these sand-sea averages.

**Sand flux**. Sand flux $\vec{q}$ (g/m/s,°) is calculated as a vector based on the 10-m wind vector $\{u_{10}, v_{10}\}$ (m/s,°) and the precipitation flux $P$ (kg/m²/s). Wind vectors used in the calculation are instantaneous 3-hourly values, used instead of means to reflect the variability in winds, and precipitation flux values are the 3-hourly average. Precipitation diminishes wind-driven sand flux by increasing the threshold wind required to move sand—so much so that rainfall essentially shuts off sand flux, through the creation of liquid bridges between grains that produce a capillary force opposing motion[8]. We parameterize this effect as any precipitation flux exceeding a very small value ($10^{-4}$ kg/m²/s or 8.64 mm/day), during the 3-hourly interval immediately preceding the instantaneous wind vector measurement, causes sand flux to be zero regardless of wind speed. We choose this parameterization for its simplicity and in lieu of a robust and numerically-efficient alternative, and do not consider the implicit role of precipitation in changing threshold via vegetation.

We have assessed the impact of this precipitation effect implementation relative to neglecting precipitation's role completely for sand flux in Fig. S8: the implementation reduces the overall sand flux (necessarily) by less than 10%, and differences in how it alters the change in flux measures across the century is negligible. Furthermore, we have implicitly assessed the importance of any higher-order precipitation effect via vegetation by looking at the vegetation mass in the sand seas—and its change—relative to the rest of the planet using the EC-Earth3 sister ESM (EC-Earth3Veg), which has an active land biosphere module, in Fig. S9. In that ESM, the sand seas all have small or zero vegetation, and changes in vegetation across the century are small or zero.

When precipitation does not play a role, the wind and sand flux are related in the following way. Sand flux direction is taken as the same direction as the 10-m wind, $\angle \vec{q} = \arctan 2(v_{10}, u_{10})$. We assume that sand flux magnitude obeys the following relationship[7,20,26],

$$|\vec{q}| = \begin{cases} 0, & u_* \le u_{*,th} \\ A \dfrac{u_{*,th}\rho_f}{g}\left(u_*^2 - u_{*,th}^2\right), & u_* > u_{*,th} \end{cases} \quad (1)$$

where $A = 5$ is a dimensionless constant of proportionality found through field calibration[7], $u_{*,th}$ (m/s) is the threshold friction velocity, $\rho_f = 1.225$ (kg/m³) is the fluid (air) density, $g = 9.8$ (m²/s) is gravity and $u_*$ (m/s) is the friction velocity. It should be noted that $\vec{q}$ is not strictly the sand flux, but instead the sand flux capacity which would occur on flat and fully-erodible sand[19].

Though not ideal as it neglects atmospheric stability effects below 10 meters, in lieu of a more robust relationship for the strongly-forced sand-sea boundary layer we assume friction velocity $u_*$ (m/s) is related to the 10-m wind speed using the Law of the Wall[7,19,20,26],

$$|\{u_{10}, v_{10}\}| = \frac{u_*}{\kappa}\ln\left(\frac{10}{z_0}\right) \quad (2)$$

where subscript '10' denotes the 10-m elevation of measurement, $\kappa = 0.4$ is von Karman's constant, and $z_0 = 10^{-3}$ (m) is the roughness length at the scale of sand transport which we assume (imperfectly) is a global constant[5]. We note that the boundary layer scheme in the EC-Earth3 does account for quasi-steady atmospheric stability effects[41].

The threshold friction velocity $u_{*,th}$ is chosen as the saltation impact threshold[26]. We choose not to include separate initiation and cessation thresholds because other sources of variability likely contribute more error: variability of friction velocity within the timestep due to turbulence[7]; effects of topographic variations on friction velocity (and the threshold itself) over the grid spacing[19] (including from the dunes themselves, foremost giant complex dunes); and variation in the threshold due to unknown locally-varying sediment characteristics. Nonetheless, our approach represents a significant improvement over most large-scale studies that omit threshold altogether[13,21,22], choosing instead to employ the so-called 'drift potential' which does not allow analysis of flux events. We parameterize the threshold using a common formula[19,26,27],

$$u_{*,th} = B\sqrt{\frac{\rho_s - \rho_f}{\rho_f}gd} \quad (3)$$

where $B = 0.082$ is a dimensionless constant of proportionality found through experimental calibration[19], $\rho_s = 2650$ (kg/m³) is the density of sand, and $d = 300$ ($\mu$m) is the grain diameter. We take all the constants to be the same across Earth

since it is not well-known what representative values should be for each sand sea in the data set.

**Incipient wavelength.** The incipient wavelength of dunes has been measured in the field and experimentally to follow the relationship[26–28],

$$\lambda_c = \frac{2\pi L_{sat}\mathcal{A}}{\mathcal{B} - \left(u_*/u_{*,th}\right)^{-2}/\mu} \tag{4}$$

where $L_{sat} = Cd\rho_s/\rho_f$ (m) is the saturation length ($C = 2.2$ is a dimensionless constant of proportionality found through experimental calibration[26]), $\mathcal{A} = 3.6$ and $\mathcal{B} = 1.9$ are dimensionless hydrodynamical constants calibrated to field data that explain the initial development of dunes through linear stability analysis[27], and $\mu = \tan(34°)$ is the friction coefficient corresponding to the angle of repose for natural sand. The other parameters are defined in the Methods section above.

## Data availability

The 3-hourly data from the EC-Earth3 ESM used in this study are available in the CMIP6 database https://esgf-node.llnl.gov/search/cmip6/ (or at another node). The sand sea GIS file generated in this study are provided in the repository https://doi.org/10.5281/zenodo.6562611. The CMIP6 data for ESM comparison used in this study are available using the Google Cloud API.

## Code availability

Code to reproduce this paper can be found at https://doi.org/10.5281/zenodo.6562611.

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

## Acknowledgements

We thank Claire Masteller for useful discussions, Gary Kocurek, Harrison Gray, David Thomas and one anonymous reviewer for their constructive reviews, and National Science Foundation funding (award NRI #1734355) to D.J.J. Acknowledgment is made to the Donors of the American Chemical Society Petroleum Research Fund for partial support of this research through grant #61536-ND8 to D.J.J. Any use of trade, firm, or product names is for descriptive purposes only and does not constitute endorsement by the U.S. government.

## Author contributions

Conceptualization, Data Curation, Formal Analysis, Methodology, Software, Validation, Visualization and Writing–original draft, A.G.; Investigation, A.G. & A.E.; Project Administration, Writing–review & editing, all authors; Resources, Funding Acquisition, Supervision, D.J.J.

## Competing interests

The authors declare no competing interests.

ARTICLE
