## [Peer Review File · Nature Communications]

21st-century stagnation in unvegetated sand-sea activityReviewers' Comments:

Reviewer #1:

Remarks to the Author:

The manuscript uses ensemble runs of the Earth System Model EC-Earth3 and the Coupled Climate Model Intercomparison Project (CMIP6) to determine the current activity level of global sand seas and then forecast the change in this activity into the remaining 21st century with scenarios of predicted climate change. The underlying premise is that dune activity is a balance between vegetation that stabilizes dunes, and winds above threshold speed that cause sand transport. The caveat here is that there is sufficient moisture to support vegetation. Additionally, although it varies geographically, wind regimes are typically seasonal in both intensity and direction, and often strongly diurnal. Moreover, sand transport events are usually intermittent, lasting hours or days, and usually separated by longer periods when the winds, if these occur at all, are below the threshold for sand transport. The potential for plant colonization of dune surfaces is a function of precipitation/soil moisture and dune surface stability. Plants are able to colonize dunes during periods of wind quiescence; with increasing plant cover, progressively greater wind energy is needed to reactivate the surface. With changes in precipitation, potential evaporation and wind energy, dune fields and sand seas evolve between end-members of a fully erodible surface of active dunes and total stabilization of the dune morphology by vegetation.

The amount of data synthesized in this manuscript is truly staggering. For the historical base, global sand seas (45 in total) were spatially defined, precipitation and the wind regimes were derived from reanalysis data, the wind regime was characterized by intensity, wind event duration, wait-times between wind events, and wind directionality. Average sand flux in the sand seas flux was calculated. Using this base, the modeled forecast uses accepted hypothetical radiative forcing resulting from greenhouse gas emission. Because of the global nature of the study and relative paucity of ground data for many sand seas, assumptions and simplifications were made. These include the coarse gridding (≤ 100 km), ignoring atmospheric stability, and assumptions of a constant surface roughness length and quartz grains of 0.3 mm. Reanalysis data are also limited in quality where ground stations are few, as is the case for sand seas. In my view, given the global nature of the study, the assumptions and data use are defensible. Models are models, and being a field sedimentologist, I take these with a grain of salt. My guess is that, all considered, the results may be wrong in detail, but the overall trends, what is of interest here, should be good.

The model results are intriguing. With few exceptions, dune activity in the global sand seas is forecast to decrease from current levels into the future 21st century. While an overall relaxing of the wind regime occurs, most of decrease results from longer wait-times between sand-transporting wind events and a decrease in winds that occur outside of the main wind season. Owing to the latter point, wind regimes become more unidirectional, thus favoring transverse dune morphologies. The authors correctly note that the limited time of a century is not sufficient to reform the large dunes that populate many sand seas, but will give rise to superimposed transverse dunes (something I would view as a pattern-reformation process). With decreased winds and longer wait times between wind events, conditions would be prone for the expansion of vegetation, raising the potential for greater stabilization of dunes by vegetation. Additionally, parabolic dunes, associated with partial stabilization, should become more common. Another implication is that the incipient wavelength of dunes emerging in the surface instability mode should increase. Changes in the wind regime are thought to be more important than predicted changes in precipitation/soil moisture in the forecast changes in sand sea activity.

To my knowledge, results here are the first forecasts for currently active sand seas based upon projections of future climate change. The extent to which the projected changes are the result of changes in the wind regime is an entirely novel finding. Previous forecasts have focused on specific areas where dunes are currently largely stabilized by vegetation. For example, in an older paper, Muhs and Maat (1993), specifically addressed the now largely vegetated and stabilized dune fields of the

Great Plains, USA, and anticipated dune reactivation based upon regional forecast of increased temperature and decreased precipitation. More recently, Thomas, Knight & Wiggs (2005 paper in Nature), addressed vegetated dune fields in South Africa via global climate models, and concluded that global warming will lead to significant dune reactivation by 2099 because of increased potential evapotranspiration and increased wind energy. Note that this is mostly consistent with a moderate increase in dune activity in southern Africa in this manuscript. It should be noted that over the Quaternary for global sand seas, Milankovitch-forced climatic change has generally resulted in wetter, vegetated and stabilized dunes during times of greater solar insolation (interglacial), and drier, windier, less vegetated and more active dunes during times of lesser solar insolation (glacial). This shows up in most of the climatic/vegetation models and proxy data. What I am unclear about is how comparable global warming by solar insolation is to warming by greenhouse gas forcing.

While the treatment of aspects the wind regime (especially Figure 3) is truly innovative, especially consideration of wait times (Figure ED2), I was left wondering about precipitation. Precipitation is mostly addressed in Figure ED6. Here and as briefly noted in the text, precipitation generally tends to increase in the forecasts (ED6-d). My question is whether precipitation, especially during wait times when vegetation is most apt to colonize dunes, is indeed sufficient for plants. Many of the sand seas are hyperarid and devoid of vegetation because of aridity, and not because the wind energy is too high. You could reduce winds to zero, and still not have vegetation stabilize dunes, unless there is sufficient water available. Dune activity can be expected to decrease with the forecast waning of winds, but my impression was that sand-sea "stagnation" largely resulted from increased vegetation? In hindcast climatic modeling, precipitation is generally taken as the main control on vegetation types within reasonably established limits (e.g., savanna, shrub, desert with less 10% plant cover). The Kutzbach et al. paper (PNAS, Feb 4, 2020, v. 117, #5, p. 2255-2264) is a good example that covers the Sahara and Arabia, including modeled plant distribution. The authors of this manuscript may be coming from a perspective of White Sands, New Mexico, where wind deceleration within the growing boundary layer is sufficient to reduce sand transport and allow for downwind colonization of the dunes by vegetation. White Sands, however, occupies a former lake bed surrounded by short-grass prairie/savanna. White Sands is not the Sahara or the Rub al' Khali. At any rate, unless I am missing something, the issue of precipitation needs to be clarified.

To address the specific reviewer questions. The text is well done, and I spent most of my time trying to unpack the figures. Each figure contains a massive amount of information, and I am sure I am still missing some of the points. The results are certainly noteworthy and original. The work would be of interest to several fields including: climate change impact upon surface environments, anthropogenic impacts, desert geomorphology, interpretation of ancient sand seas with climatic forcing, potential population migration with climatic change. The methodology is laid out and multiple sources are given to reproduce the data. For those of us chiefly concerned with dune geomorphology, there is a wealth of data here that I hope I can unpack.

Gary Kocurek
Geoscience, UT-Austin

Reviewer #2:
Remarks to the Author:

There is a lot of merit in this manuscript and the underlying analysis, thanks for carrying this out! I've noted below some specific points through the manuscript and some broader points. I think addressing these would greatly enhance things as they stand. There is an especial need to tighten the early paragraphs, and I raise a fundamental issue with the dune morphology analysis.

Detailed points

Abstract and line 11 etc. A lot of sand sea areas do not have dunes- they may be sand sheets, deposits of ephemeral or palaeolakes etc within. This I would argue is the norm (whether the Namib, Arabia, Inner Mongolia etc. I would suggest a slight reworking to be factually correct.

Line 9 – again, to be correct, can you evidence that they are generally windier than other earth regions? Wind energy may be reduced in some/places since primary forms were built- a trait we suggest is very relevant in e.g. even the part of the Kalahari that has evidence of modern dune activity.

Given the paper makes an important distinction between active and inactive sand seas (e.g. in abstract and primary focus), these opening lines do need revision to be clear that not all sand seas are active, barren, erodible today- hence the common but in reality, over-simple distinction between active and relict or inactive sand seas. This first para sets a stage that in effect is then contradicted by the start of para 2 (line 28). There are some simple, but critical, fixes to make to this opening paragraph.

Line 16 'since the threshold condition is frequently met the landscape is persistently in a near-critical condition': I'd debate this statement and its generality, not just through the points above, but by the strong seasonality experienced in many sand seas. In the Namib e.g., that critically is met on a relatively limited number of days, and is distinctly seasonal.

Line 24- some distinct sand seas yes, but principal dust sources globally are well evidenced as dry lake beds. Text needs nuancing.

Line 32: must say active sand seas, and in line 33.

Great that the focus is on active sand seas, but the paper would be stronger by at least greater reference to possible alternative trends in currently inactive sand seas yes, you note some different predictions for these, but this should still be flagged.

Good modelling and data set choice for M1.

I'm surprised that M3 chooses such a simple parameterisation of precip/wind interactions on sand flux, since veg is such a critical intermediary (and can be in even 'active sand seas' for significant runs of years). What estimates/tests were made on the impact of this decision? Could have very substantial consequences for the model outcomes? I would like this more explicitly considered and it needs mentioning in the paper core not just the methods section.

Lines 58 etc- haven't quite got my head around the effects of the 5 year smoothing on flux seasonality, but as flux seasonality is such a reality in field gathered empirical data, I am a bit worried about this, though you do discuss in the next paragraph.

Line 88: the hemispheric contrast is significant. It may warrant better exposure in the paper's first para as is a missed opportunity to detail the model outcomes in a very useful way.

Personal declared interest- like the link to the regional back up with our old Kalahari work- here you draw a link between active Namib and largely stable Kalahari- which to my mind adds further justification for some clarification and better text in the opening part of the paper (comments above).

Line 92- extreme event inc in S. Af mirrors many other modelling studies of climate trends- so good verification.

Line 96-7: not clear what you mean by 'to know dune morphology'? Grammar issue.

Line 105 on- yes, sure small dunes grow into big dunes, but as you sort of note earlier in the paragraph, existing big dunes don't disappear/reset over night/over a century. So I am not convinced that you can assess changes in wavelength of incipient dunes, as in many of the major sand seas big dunes are ancient and have lasted through many climate change regime changes in the Quaternary, and the smaller forms are part of the growth of the bigger forms (as we note in work from UAE,

Atkinson Thomas etc al etc) or they ride on as parts of compound or complex dunes. Therefore, the feedback of the big forms on the boundary airflow field is a key part of what happens to the incipient/smaller forms, and is covered in the literature. I need a bit of convincing that the theoretical analysis in this section has an evidence base for its realworld implications, because you are not starting with a flat sand bed. You cite the Namib, and its large dunes: you don't reference at all that these are almost exclusively compound/complex forms with the smaller dune orientations clearly impacted on by the boundary layer intrusion effects of the large forms on air flow. There is good material in this section, and in Figure 3, but it does not tie in enough of the real variables from the field that have huge implications for what is modelled.

Line 173, yes to dust budget reduction, but in the contexts whether the dunes generate dust (see earlier point) you miss noting the potential, likely significant impact, on carbon draw down if these sand seas get more vegetated, if the modelled findings hold.

Line 181: not perhaps as at odds as suggested, especially in extreme events (which inc droughts) rise (eg in sn Africa as indeed they are along with wet events). I don't think you can simply compare the effects on un veg sand seas with those oin currently veg sand seas. E.g, the eastern Namibia Kalahari dunes were hugely devegetated, and active, during the sequence of drought years up to Dec 2021 when extreme rainfall occurred. Think there is scope to nuance the interpretation here as there is actually something potentially more interesting that what is currently written.

Reviewer #3:

Remarks to the Author:

This is an excellent and novel study that looks at the potential of active sand seas to stay active in the future. It covers an extensive number of dune fields at different sizes and spread all over the world. It takes a fresh and unique approach to look at activity via flux rather than dune mobility indices that are more commonly considered. It will be of interest to a broad audience, both from the geomorphology and ecology community as well as climate change audiences. There are a couple of areas that might warrant a bit more discussion or investigation, but these are quite minor. First, I think the authors might consider explaining/justifying the precipitation threshold better, or just remove it. Did the use of this threshold and the amount and number of hours since rain (line 318) have much of an impact on the actual flux? Second, is there a better way of ordering the deserts to better tease apart any similarities? At the moment Table ED1 and the figure orders are by desert size, but I didn't see any specific relationship you might expect from desert size. Perhaps grouping by current precip/aridity, windiness, or some other attribute might help here. Third, I think it needs to be made clearer that this does not account for precipitation or seed requirements to actually facilitate plant growth, as I suspect you would still need these to grow vegetation even if the dunes decreased their mobility.

Authors' response to Reviewer 1:

The manuscript uses ensemble runs of the Earth System Model EC-Earth3 and the Coupled Climate Model Intercomparison Project (CMIP6) to determine the current activity level of global sand seas and then forecast the change in this activity into the remaining 21st century with scenarios of predicted climate change. The underlying premise is that dune activity is a balance between vegetation that stabilizes dunes, and winds above threshold speed that cause sand transport. The caveat here is that there is sufficient moisture to support vegetation. Additionally, although it varies geographically, wind regimes are typically seasonal in both intensity and direction, and often strongly diurnal. Moreover, sand transport events are usually intermittent, lasting hours or days, and usually separated by longer periods when the winds, if these occur at all, are below the threshold for sand transport. The potential for plant colonization of dune surfaces is a function of precipitation/soil moisture and dune surface stability. Plants are able to colonize dunes during periods of wind quiescence; with increasing plant cover, progressively greater wind energy is needed to reactivate the surface. With changes in precipitation, potential evaporation and wind energy, dune fields and sand seas evolve between end-members of a fully erodible surface of active dunes and total stabilization of the dune morphology by vegetation.

This is an excellent summary of the role of state-dependent evolution of a system where dunes and vegetation compete. In the present manuscript we limit the scope of our study areas to those that are presently unvegetated, and do so because we acknowledge that quantifying the interplay outlined by Gary above (i.e. magnitude of vegetation, its rate of change, and its role in altering wind energy transfer to the bed) is a complex task. We also add that the issue of how presently unvegetated arid regions would evolve has received less research attention generally.

The amount of data synthesized in this manuscript is truly staggering. For the historical base, global sand seas (45 in total) were spatially defined, precipitation and the wind regimes were derived from reanalysis data, the wind regime was characterized by intensity, wind event duration, wait-times between wind events, and wind directionality. Average sand flux in the sand seas flux was calculated. Using this base, the modeled forecast uses accepted hypothetical radiative forcing resulting from greenhouse gas emission. Because of the global nature of the study and relative paucity of ground data for many sand seas, assumptions and simplifications were made. These include the coarse gridding (≤ 100 km), ignoring atmospheric stability, and assumptions of a constant surface roughness length and quartz grains of 0.3 mm. Reanalysis data are also limited in quality where ground stations are few, as is the case for sand seas. In my view, given the global nature of the study, the assumptions and data use are defensible. Models are models, and being a field sedimentologist, I take these with a grain of salt. My guess is that, all considered, the results may be wrong in detail, but the overall trends, what is of interest here, should be good.

We are pleased to see that the reviewer is 'on the same page' as us with respect to the assumptions we are sadly forced to make about the details of each individual site. Atmospheric stability is accounted for in the model but as is pointed out above, in some locations this *may* be wrong in detail but there are no observations to test against (since they are notoriously sparse in expansive sand seas).

The model results are intriguing. With few exceptions, dune activity in the global sand seas is forecast to decrease from current levels into the future 21st century. While an overall relaxing of the wind regime occurs, most of decrease results from longer wait-times between sand-transporting wind events and a decrease in winds that occur outside of the main wind season. Owing to the latter point, wind regimes become more unidirectional, thus favoring transverse dune morphologies. The authors correctly note that the limited time of a century is not sufficient to reform the large dunes that populate many sand seas, but will give rise to superimposed transverse dunes (something I would view as a pattern-reformation process). With decreased winds and longer wait times between wind events, conditions would be prone for the expansion of vegetation, raising the potential for greater stabilization of dunes by vegetation. Additionally, parabolic dunes, associated with partial stabilization, should become more common. Another implication is that the incipient wavelength of dunes emerging in the surface instability mode should increase. Changes in the wind regime are thought to be more important than predicted changes in precipitation/soil moisture in the forecast changes in sand sea activity.

We're very glad that the manuscript has conveyed our main results to this expert reviewer clearly! The reviewer's phrase 'pattern-reformation process' is a nice framing of the morphological cannibalization of the dune field, cascading upward from the smallest scale; we have included it in the manuscript on line 126.

To my knowledge, results here are the first forecasts for currently active sand seas based upon projections of future climate change. The extent to which the projected changes are the result of changes in the wind regime is an entirely novel finding. Previous forecasts have focused on specific areas where dunes are currently largely stabilized by vegetation. For example, in an older paper, Muhs and Maat (1993), specifically addressed the now largely vegetated and stabilized dune fields of the Great Plains, USA, and anticipated dune reactivation based upon regional forecast of increased temperature and decreased precipitation. More recently, Thomas, Knight & Wiggs (2005 paper in Nature), addressed vegetated dune fields in South Africa via global climate models, and concluded that global warming will lead to significant dune reactivation by 2099 because of increased potential evapotranspiration and increased wind energy. Note that this is mostly consistent with a moderate increase in dune activity in southern Africa in this manuscript. It should be noted that over the Quaternary for global sand seas, Milankovitch-forced climatic change has generally resulted in wetter, vegetated and stabilized dunes during times of greater solar insolation (interglacial), and drier, windier, less vegetated and more active dunes during times of lesser solar insolation (glacial). This shows up in most of the climatic/vegetation models and proxy data. What I am unclear about is how comparable global warming by solar insolation is to warming by greenhouse gas forcing.

We agree with the reviewer on the points above and note to the editor that we had included the two references discussed in the original submission. While we would also be eager to understand the paleoclimate similarities, we are not certain that the role of Milankovitch-cycle and anthropogenic climate change for sand sea activity are equivalent, so we have decided to omit any discussion of this point: it is not central to the results of the paper, and we know there are not paleoclimate simulations of the equivalent fidelity using the same Earth System Model in order to test it.

While the treatment of aspects the wind regime (especially Figure 3) is truly innovative, especially consideration of wait times (Figure ED2), I was left wondering about precipitation. Precipitation is mostly addressed in Figure ED6. Here and as briefly noted in the text, precipitation generally tends to increase in the forecasts (ED6-d). My question is whether precipitation, especially during wait times when vegetation is most apt to colonize dunes, is indeed sufficient for plants. Many of the sand seas are hyperarid and devoid of vegetation because of aridity, and not because the wind energy is too high. You could reduce winds to zero, and still not have vegetation stabilize dunes, unless there is sufficient water available. Dune activity can be expected to decrease with the forecast waning of winds, but my impression was that sand-sea “stagnation” largely resulted from increased vegetation? In hindcast climatic modeling, precipitation is generally taken as the main control on vegetation types within reasonably established limits (e.g., savanna, shrub, desert with less 10% plant cover). The Kutzbach et al. paper (PNAS, Feb 4, 2020, v. 117, #5, p. 2255-2264) is a good example that covers the Sahara and Arabia, including modeled plant distribution. The authors of this manuscript may be coming from a perspective of White Sands, New Mexico, where wind deceleration within the growing boundary layer is sufficient to reduce sand transport and allow for downwind colonization of the dunes by vegetation. White Sands, however, occupies a former lake bed surrounded by short-grass prairie/savanna. White Sands is not the Sahara or the Rub al' Khali. At any rate, unless I am missing something, the issue of precipitation needs to be clarified.

This is a fair comment from the reviewer. We had originally centered the idea that vegetation will change if sand transport does, since the geomorphology-focused community view is that there is an active competition between the conservative role of vegetation and active surface processes (e.g. Yizhaq et al., 2007; Duran & Hermann, 2006). This is not necessarily the case, especially when precipitation is so infrequent and/or low that it increasing it moderately would still not be sufficient for vegetation. We have added the sentence “However, in many of Earth's hyper-arid landscapes, including some of the 45 sand seas studied here, the principle bottleneck for the rise of vegetation (and therefore parabolic dunes) is not insurmountable wind power but a lack precipitation” to the Discussion (Lines 180-182) to make this important point clear. Furthermore, along with comments raised by other reviewers, we endeavored to quantify the change in vegetation in the sand seas using the sister model (EC-Earth3Veg) to the ESM employed in the main text (EC-Earth3). In a new supplementary figure (Figure S8), we show that vegetation in the sand seas is zero and remains zero in the future under the most severe future scenario for 28 of the 45 sand seas, and the others change modestly but all have low vegetation mass relative to global land. We explain this in the figure caption and in the Methods M3 section (Lines 242-250).

To address the specific reviewer questions. The text is well done, and I spent most of my time trying to unpack the figures. Each figure contains a massive amount of information, and I am sure I am still missing some of the points. The results are certainly noteworthy and original. The work would be of interest to several fields including: climate change impact upon surface environments, anthropogenic impacts, desert geomorphology, interpretation of ancient sand seas with climatic forcing, potential population migration with climatic change. The methodology is laid out and multiple sources are given to reproduce the data. For those of us chiefly concerned with dune geomorphology, there is a wealth of data here that I hope I can unpack.

Gary Kocurek

Geoscience, UT-Austin

Thank you for these constructive, positive comments!

Authors' response to Reviewer 2:

There is a lot of merit in this manuscript and the underlying analysis, thanks for carrying this out! I've noted below some specific points through the manuscript and some broader points. I think addressing these would greatly enhance things as they stand. There is an especial need to tighten the early paragraphs, and I raise a fundamental issue with the dune morphology analysis.

Abstract and line 11 etc. A lot of sand sea areas do not have dunes- they may be sand sheets, deposits of ephemeral or palaeolakes etc within. This I would argue is the norm (whether the Namib, Arabia, Inner Mongolia etc.) I would suggest a slight reworking to be factually correct.

Thank you for these comments. We have reworded this introductory material to make clearer the range of geomorphic conditions in these regions. The abstract now mentions sand seas as "containing large areas of aeolian dunes", and in the introductory paragraph we added more nuanced phrasing such as "regions with dunes" and "the parts of these landscapes dominated by loose sand". We changed "everywhere" to "wherever unconsolidated sediment occurs" (first paragraph of introductory text).

Line 9 – again, to be correct, can you [provide] evidence that they are generally windier than other earth regions? Wind energy may be reduced in some/places since primary forms were built- a trait we suggest is very relevant in e.g. even the part of the Kalahari that has evidence of modern dune activity.

Given the paper makes an important distinction between active and inactive sand seas (e.g. in abstract and primary focus), these opening lines do need revision to be clear that not all sand seas are active, barren, erodible today- hence the common but in reality, over-simple distinction between active and relict or inactive sand seas. This first para sets a stage that in effect is then contradicted by the start of para 2 (line 28). There are some simple, but critical, fixes to make to this opening paragraph.

Line 16 'since the threshold condition is frequently met the landscape is persistently in a near-critical condition': I'd debate this statement and its generality, not just through the points above, but by the strong seasonality experienced in many sand seas. In the Namib e.g., that critically is met on a relatively limited number of days, and is distinctly seasonal.

True. This text was meant as general context statements, but to be accurate we have added the qualifier, "at least on seasonal or shorter time scales" after "frequently".

Line 24- some distinct sand seas yes, but principal dust sources globally are well evidenced as dry lake beds. Text needs nuancing.

Yes, we agree and thanks for pointing this out. We have deleted the phrase that this comment was based on (about the global dust budget) and just left it as describing a tipping point for an arid landscape (whether vegetation colonizes or not).

Line 32: must say active sand seas, and in line 33.

Thank you, we added "active" there.

Great that the focus is on active sand seas, but the paper would be stronger by at least greater reference to possible alternative trends in currently inactive sand seas yes, you note some different predictions for these, but this should still be flagged.

This is a fair point. We have included a note in the Discussion (Lines and raise the pronounced increase in predicted wind speed over the Thar Desert seen in Figure 1a there too: "While we do not focus on those cases, we note that the wind strength changes seen in the ESM around the partially-vegetated Thar Desert, and coastal dunes in north Chile and south Peru, warrant further study of potential short-term reactivation (Fig. 1a)."

Good modelling and data set choice for M1.

Thank you!

I'm surprised that M3 chooses such a simple parameterisation of precip/wind interactions on sand flux, since veg is such a critical intermediary (and can be in even 'active sand seas' for significant runs of years). What estimates/tests were made on the impact of this decision? Could have very substantial consequences for the model outcomes? I would like this more explicitly considered and it needs mentioning in the paper core not just the methods section.

We agree that vegetation is sometimes a critical intermediary between the precipitation rate and sand transport in arid environments. Implementing such a scheme, where the threshold wind speed required to move sand, and

the sensitivity of transport above that threshold to wind in-excess of that threshold, is modulated by vegetation, which is in turn a function of past precipitation, is beyond the scope of this work. Biospheric modules of Earth System Models are designed by teams of researchers for this very purpose: it is an active area of research that we are unfortunately experts in. That is why we specifically decided to limit our study's scope to presently unvegetated dune fields and the present century, so as to limit the influence of neglecting the potential impact of vegetation changing due to precipitation. We decided to go down the route of only allowing precipitation to influence threshold within the 3-hour period (one model time-step) it occurred, since this is the first-order role of precipitation for sand transport in unvegetated environments. Due to computational constraints, implementing a scheme where the threshold relies on a memory of many previous time-steps of model data is unfortunately not available to us, and it is not clear exactly what functional form one would use to do this, let alone how to effectively parameterize this globally. Two tests have been performed to estimate the impact of this decision.

First, we removed the role of precipitation in our model to assess what the first-order parameterization does as opposed to the zeroth-order parameterization (null model; ignore it). We expect that any higher-order parameterizations of precipitation via vegetation would have smaller effects than this within a century. As an additional supplementary information figure in the revised submission is a comparison of the results with and without the precipitation effect. One can see that the magnitude and directions of changes to the variables is very similar for all dune fields almost across the board. Necessarily, the total magnitude of fluxes are greater without the precipitation effect, however these changes are moderate given the aridity of sand seas. We have explained this in the Figure S8 caption and added the lines to the Methods M3 section: "We have assessed the impact of this precipitation effect implementation relative to neglecting precipitation's role completely for sand flux in Figure S8: the implementation reduces the overall sand flux (necessarily) by less than 10%, and differences in how it alters the change in flux measures across the century is negligible." (Lines 242-245).

Second, we look at the changes and magnitudes in total vegetation carbon mass per area on the sand seas using the EC-Earth3's counterpart ESM that actually includes a biospheric module, the EC-Earth3-Veg. We did not use this model in our main analysis because it does not have the timestep fidelity (monthly data only) we require for a threshold-based implementation of the sand transport (Methods M3). We note before diving into this model's results, that in Figure S7 one can see that the changes in winds in all the sand seas are in agreement between the two ESM flavors, implying that accounting for vegetation at the planet-scale has no significant implicit influence on winds that move sand. Two things to note with this result: 1) the total vegetation carbon mass per area, regardless of scenario (either axes on the main panel), in the sand seas is small relative to the distribution of values for all of land (inset)—in fact, the majority (28 of 45) of the sand seas are simulated to have zero vegetation in the EC-Earth3 throughout the year; and 2) the changes between the historical average and end-of-century most severe SSP (5-8.5) scenario case are also small—the 28 without vegetation don't change from zero, and the others remain small. Interestingly, vegetation is predicted to increase in 14 of the 17 sand seas with non-zero predicted vegetation carbon mass. We have included this as an additional supplementary information figure in the revised submission (Figure S9). The caption explains what is written above, added we added lines to the Methods M3 section: "Furthermore, we have implicitly assessed the importance of any higher-order precipitation effect via vegetation by looking at the vegetation mass in the sand seas---and its change---relative to the rest of the planet using the EC-Earth3 sister ESM (EC-Earth3Veg), which has an active land biosphere module, in Figure S9. In that ESM, the sand seas all have small or zero vegetation, and changes in vegetation across the century are small or zero." (Lines 245-250).

Lines 58 etc- haven't quite got my head around the effects of the 5 year smoothing on flux seasonality, but as flux seasonality is such a reality in field gathered empirical data, I am a bit worried about this, though you do discuss in the next paragraph.

Our smoothing method doesn't affect any seasonal bias here: the smoothed lines in Figure 2a are pretty smooth because a) we are averaging over multiple ensemble members (actualizations of the climate system where there might be one member that has a very windy summer but the rest have normal ones) and b) we have sand seas across the planet, and indeed on the separate hemispheres, so the seasonal signal from any given sand sea is damped by the other sand sea signals.

Line 88: the hemispheric contrast is significant. It may warrant better exposure in the paper's first para as is a missed opportunity to detail the model outcomes in a very useful way.

Personal declared interest- like the link to the regional back up with our old Kalahari work- here you draw a link between active Namib and largely stable Kalahari- which to my mind adds further justification for some clarification and better text in the opening part of the paper (comments above).

Line 92- extreme event inc in Sn Af mirrors many other modelling studies of climate trends- so good verification.

We have made special reference to the reviewer's work (assuming they are talking about the Thomas et al., 2005 paper) in the introductory paragraph "(i.e. the Kalahari (Thomas et al., 2005))" in Line 52 as requested. We highlight the text in the final paragraph of the first Results subsection which already states much of what the reviewer raised here, which we think is a sufficient discussion: "The southern hemisphere sand seas in central Australia and southern Africa instead see a moderate increase in activity which is qualitatively consistent with previous studies[13,21,22] (Fig. S3a). Despite this hemispheric contrast..."

Line 96-7: not clear what you mean by 'to know dune morphology? Grammar issue.

We apologize that this was not clearly phrased before. It's been changed now to "is not sufficient to determine dune morphology."

Line 105 on- yes, sure small dunes grow into big dunes, but as you sort of note earlier in the paragraph, existing big dunes don't disappear/reset over night/over a century. So I am not convinced that you can assess changes in wavelength of incipient dunes, as in many of the major sand seas big dunes are ancient and have lasted through many climate change regime changes in the Quaternary, and the smaller forms are part of the growth of the bigger forms (as we note in work from UAE, Atkinson Thomas etc al etc) or they ride on as parts of compound or complex dunes. Therefore, the feedback of the big forms on the boundary airflow field is a key part of what happens to the incipient/smaller forms, and is covered in the literature. I need a bit of convincing that the theoretical analysis in this section has an evidence base for its realworld implications, because you are not starting with a flat sand bed. You cite the Namib, and its large dunes: you don't reference at all that these are almost exclusively compound/complex forms with the smaller dune orientations clearly impacted on by the boundary layer intrusion effects of the large forms on air flow. There is good material in this section, and in Figure 3, but it does not tie in enough of the real variables from the field that have huge implications for what is modelled.

The points the reviewer raise here are all fair. We agree that local flow modification by the largest-scale dunes on the landscape can impact the smaller bedform orientations. Our results about incipient bedforms are, however, independent of orientation: they are a direct result in changes to the average above-threshold wind speeds. We think the reviewer's comments are consistent with what is written in the manuscript: incipient dunes are by definition the smallest and youngest dunes on the landscape, the largest dunes are the last to change, they are centuries old (Lines 120-126). We note also that the theory we draw on for the incipient dune wavelength is directly validated on the back of the largest dune in that dune field (Gadal et al., 2020). Further we noted in the sand flux methods section that local flow direction can be changed by local sub-gridscale topography. We have added a direct reference to giant dunes themselves here: "(including from the dunes themselves, foremost giant complex dunes)" Line 262-263.

Line 173, yes to dust budget reduction, but in the contexts whether the dunes generate dust (see earlier point) you miss noting the potential, likely significant impact, on carbon draw down if these sand seas get more vegetated, if the modelled findings hold.

We appreciate the comment, but desert vegetation (even if increasing from the near-zero levels currently growing in the world's large sand seas) stores relatively low carbon stocks, e.g., compared to forests, due to the relatively lower biomass in sparse vegetation that tends to colonize dunes in arid regions, compared to that of trees. Nevertheless, the reviewer is correct that this is still worth mentioning. In response, we have added the line "Increased vegetation would affect the regional carbon cycle and potentially increase atmospheric CO2 drawdown, though likely to a modest degree." Developing a potential carbon budget for this situation seems beyond the scope of our investigation, and we suspect the effect would be minor compared to expansion of savanna or forest ecosystems, but perhaps could be an avenue for future study.

Line 181: not perhaps as at odds as suggested, especially in extreme events (which inc droughts) rise (eg in sn Africa as indeed they are along with wet events). I don't think you can simply compare the effects on un veg sand seas with those in currently veg sand seas. E.g, the eastern Namibia Kalahari dunes were hugely devegetated, and active, during the sequence of drought years up to Dec 2021 when extreme rainfall occurred. Think there is scope to nuance the interpretation here as there is actually something potentially more interesting that what is currently written.

This is a reasonable comment. We have changed "at odds" to "inconsistent at the global scale" in this sentence.

Authors' response to Reviewer 3:

This is an excellent and novel study that looks at the potential of active sand seas to stay active in the future. It covers an extensive number of dune fields at different sizes and spread all over the world. It takes a fresh and unique approach to look at activity via flux rather than dune mobility indices that are more commonly considered. It will be of interest to a broad audience, both from the geomorphology and ecology community as well as climate change audiences. There are a couple of areas that might warrant a bit more discussion or investigation, but these are quite minor.

We thank the reviewer for their very supportive comments!

First, I think the authors might consider explaining/justifying the precipitation threshold better, or just remove it. Did the use of this threshold and the amount and number of hours since rain (line 318) have much of an impact on the actual flux?

We have added another supplementary figure (Fig. S8) which does exactly this, and included more information about its effect/justification in the Methods section M3. We find that removing the precipitation threshold increased sand flux by about 8.5% overall, meaning that it has a non-negligible influence on the magnitude of transport. We also found, though, that the threshold does little to affect the changes in the flux over the century. We're glad the reviewer raised this, have included the additional check in the supplementary for completeness, and have kept the precipitation effect in the main results.

Second, is there a better way of ordering the deserts to better tease apart any similarities? At the moment Table ED1 and the figure orders are by desert size, but I didn't see any specific relationship you might expect from desert size. Perhaps grouping by current precip/aridity, windiness, or some other attribute might help here.

We ordered the sand seas by size since the global average sand flux is weighted by size. We appreciate what the reviewer is after though, and point to the text where we delve into the regional effects in the Results section (Line 100-111, 145, 149-153), the modified text in the Discussion which expands on regional effects (Lines 184-188), and the Figures 1 and S4 which do much to show the windiness and change in windiness spatially across the globe. We do not group by any single parameter like this because the regional climates for the sand seas change differently irrespective of a single parameter, which can be seen in Figure S3.

Third, I think it needs to be made clearer that this does not account for precipitation or seed requirements to actually facilitate plant growth, as I suspect you would still need these to grow vegetation even if the dunes decreased their mobility.

This is a very fair point, also raised by Gary Kocurek above, and we have introduced another supplementary figure (Fig. S9) which shows that the vegetation in the sand seas is sparse, and does not change much into the future when it is subject to changing climate over the next century (as predicted by the sister model of the one employed in the main manuscript that *does* have an active vegetation module). We have noted these results in the figure caption and in the Methods section M3.

Reviewers' Comments:

Reviewer #1:

Remarks to the Author:

I read the revised manuscript and the authors' response to the reviews. I was impressed by the depth of the responses, the careful weighing, and the appropriate revisions. I believe that this manuscript should move forward to publication.

Reviewer #2:

Remarks to the Author:

Thanks for your strong and helpful responses to reviewer comments. I am content that you have in the main addressed well the points that I raised, and that other reviewers raised too.

Given some of the comments and acknowledgment that the study focusses not on all sand seas, and omits arguably the most sensitive vegetated systems, I suggest a revision to the title that is subtle but which avoids overstating the case made:

'21st-century stagnation in regional sand sea activity'

I am not a fan of grandstanding titles that do not convey the content of a paper most accurately.

Thank you for conducting this work: I look forward to seeing it in press.

David Thomas

Reviewer #3:

Remarks to the Author:

Overall I am satisfied with these corrections. The authors have done a great job at addressing my initial minor comments.

Authors' response to Review
Nature Communications manuscript NCOMMS-21-06748A

Authors' response to Editor:

Dear Professor Jerolmack,

Your manuscript entitled "21st-century stagnation in sand-sea activity" has now been seen again by our referees, whose comments appear below. In light of their advice I am delighted to say that we are happy, in principle, to publish a suitably revised version in Nature Communications under the open access CC BY license (Creative Commons Attribution 4.0 International License).

We therefore invite you to revise your paper one last time to address the remaining concerns of our reviewers and our editorial requests in the attached document(s). At the same time we ask that you edit your manuscript to comply with our policies and formatting requirements and to maximise the accessibility and therefore the impact of your work.

Best regards,

Dr Kasey Bolles
Associate Editor
Nature Communications

Dear Dr. Bolles,

Thank you for handling our manuscript.

We are very pleased to see the manuscript accepted at *Nature Communications*. We have made some minor changes to the manuscript to comply with the journal's policies, and have changed the title to suit the sole comment of the reviewers.

This document is structured as a response to each Reviewer's summary, and then short responses to each succeeding comment. We have attached both a 'tracked changes' version and an 'incorporated changes' version of the manuscript files to this response. In this document, the Reviewer's comments are pasted with indents and green highlight for clarity (like yours above). References to any changes we made are highlighted in blue, and the changes themselves are given below in highlighted pink with line numbers corresponding to the 'tracked changes' version of the manuscript.

We greatly appreciate your time with this manuscript and look forward to seeing it in the journal soon.

Yours sincerely,

Andrew Gunn, Amy East and Douglas Jerolmack

Authors' response to Reviewer 1:

I read the revised manuscript and the authors' response to the reviews. I was impressed by the depth of the responses, the careful weighing, and the appropriate revisions. I believe that this manuscript should move forward to publication.

We thank Reviewer 1 for their very useful and positive comments.

Authors' response to Reviewer 2:

Thanks for your strong and helpful responses to reviewer comments. I am content that you have in the main addressed well the points that I raised, and that other reviewers raised too.

Given some of the comments and acknowledgment that the study focusses not on all sand seas, and omits arguably the most sensitive vegetated systems, I suggest a revision to the title that is subtle but which avoids overstating the case made:

'21st-century stagnation in regional sand sea activity'

I am not a fan of grandstanding titles that do not convey the content of a paper most accurately.

Thank you for conducting this work: I look forward to seeing it in press.

David Thomas

We thank David for his review, particularly for the aspects about vegetation and comparison with his previous work. We have changed the title to include "vegetation" instead of "regional" since his point here appears to be mostly about narrowing its scope to be about unvegetated sand seas (in-line with the actual study). This is a fair point and we have made the change, but preferred not to use "regional" since it is a global study. We hope the editor agrees that our change addresses the comment.

Authors' response to Reviewer 3:

Overall I am satisfied with these corrections. The authors have done a great job at addressing my initial minor comments.

We thank Reviewer 3 for their initial comments and their positive remarks.

Authors' additional changes:

Added the new named reviewer to the acknowledgments

Gave the code repository a DOI

Changed the formatting of the methods section slightly to suit the journal requirements